# VocSegMRI: Multimodal Learning for Precise Vocal Tract Segmentation in Real-time MRI

**Daiqi Liu**[1]   DAIQI.DEUTSCHFAU.LIU@FAU.DE

**Johannes Enk**[1]   ENK.JOHANNES98@GMAIL.COM

**Maureen Stone**[2]   MSTONE@UMARYLAND.EDU

**Fangxu Xing**[3]   FXING1@MGH.HARVARD.EDU

**Tomás Arias-Vergara**[1,4]   TOMAS.ARIAS@FAU.DE

**Jerry L. Prince**[5]   PRINCE@JHU.EDU

**Jana Hutter**[6]   HUTTER@TNT.UNI-HANNOVER.DE

**Jonghye Woo**[3]   JWOO@MGH.HARVARD.EDU

**Andreas Maier**[1]   ANDREAS.MAIER@FAU.DE

**Paula A. Pérez-Toro**[1,4]   PAULA.ANDREA.PEREZ@FAU.DE

[1] *Friedrich-Alexander-Universität Erlangen-Nürnberg*   [2] *University of Maryland School of Dentistry*

[3] *Harvard Medical School/Massachusetts General Hospital*   [4] *Universidad de Antioquia UdeA*

[5] *Johns Hopkins University*   [6] *Leibniz University Hannover*

## Abstract

Accurate segmentation of articulatory structures in real-time MRI (rtMRI) remains challenging, as existing methods rely primarily on visual cues and overlook complementary information from synchronized speech signals. We propose VocSegMRI, a multimodal framework integrating video, audio, and phonological inputs via cross-attention fusion and a contrastive learning objective that improves cross-modal alignment and segmentation precision. Evaluated on USC-75 and further validated via zero-shot transfer on USC-TIMIT, VocSegMRI outperforms unimodal and multimodal baselines, with ablations confirming the contribution of each component.

**Keywords:** Segmentation, Multimodal Learning, Real-time MRI, Vocal Tract

## 1. Introduction

Real-time MRI (rtMRI) enables non-invasive imaging of the entire vocal tract during continuous speech, supporting both phonetic research and clinical applications such as pre-surgical planning in glossectomy and articulatory monitoring in Parkinson's disease (Toutios et al., 2016; Lammert et al., 2016; Hagedorn et al., 2014). Accurate segmentation of vocal tract structures remains challenging; while deep learning approaches, including FCN-based methods (Somandepalli et al., 2017), U-Net variants (Ronneberger et al., 2015; Ruthven et al., 2021), and foundation models such as nnU-Net (Isensee et al., 2021) and SAM (Kirillov et al., 2023), have improved performance, they rely exclusively on visual cues and ignore the rich articulatory information carried by synchronized speech. Although a recent multimodal approach integrating acoustic features with rtMRI shows promise (Jain et al., 2024), the potential of phonological features and cross-modal contrastive alignment remains unexplored. We propose **VocSegMRI**, a multimodal framework that incorporates cross-attention fusion of visual, acoustic, and phonological streams alongside a dual-level contrastive learning objective, demonstrating superior performance over unimodal and multimodal baselines on the USC-75 (Lim et al., 2021) and USC-TIMIT (Narayanan et al., 2014) datasets.

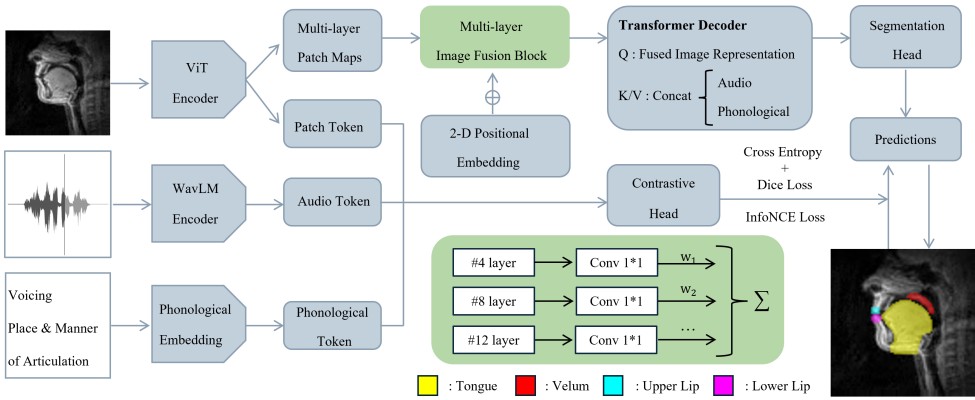

Figure 1: Overview of the proposed VocSegMRI model.

## 2. Materials & Methods

**Datasets.** We utilized data from five participants from the USC-75 dataset: one participant (538 frames) was held out as an unseen test speaker; the remaining four (1,863 frames) were used for leave-one-speaker-out cross-validation. For cross-dataset generalization, two held-out speakers (517 frames) from the USC-TIMIT dataset were used for zero-shot evaluation. All videos were downsampled to 15 fps, resized to $224 \times 224$ pixels, and normalized to $[0, 1]$.

**Method.** The overall framework of VocSegMRI is illustrated in Fig. 1. A pretrained ViT encoder extracts multi-layer patch representations from rtMRI frames (Wu et al., 2020), while synchronized audio is encoded by a pretrained WavLM model (Chen et al., 2022), and phonological features (voicing, place, and manner of articulation) are embedded via a lightweight MLP. Audio and phonological tokens are concatenated and serve as keys and values in a Transformer decoder (Vaswani et al., 2017), where fused image representations act as queries. A contrastive head projects image and audio–phonological tokens into a shared latent space, supervising the visual encoder to learn more discriminative features (Liu et al., 2025). All experiments are implemented in PyTorch 2.5.1 (Paszke et al., 2019) on a NVIDIA RTX A100 GPU, with a batch size of 12 and 40 epochs using AdamW (weight decay $1 \times 10^{-2}$). The ViT encoder is frozen for the first 15 epochs and gradually unfrozen thereafter, while the WavLM encoder remains frozen. The final loss combines cross-entropy, Dice, and contrastive terms.

## 3. Results

Results are summarized in Tab. 1. VocSegMRI outperforms both the best unimodal baseline ViT (Dice: 0.85 vs. 0.75) and the multimodal baseline Jain (Jain et al., 2024) (Dice: 0.85 vs. 0.80), achieving an ASSD of 3.17 mm. Ablation results confirm the individual contributions of cross-attention and contrastive learning. Under zero-shot transfer to USC-TIMIT, VocSegMRI maintains strong performance. Qualitative results in Fig. 2 further show that VocSegMRI produces more accurate segmentation of the velum, where other methods exhibit notable localization and boundary errors.

Table 1: Segmentation performance on USC-75 (in-domain) and USC-TIMIT (zero-shot). The best results are highlighted in **bold**.

| Methods | USC-75 | | USC-TIMIT | |
|---|---|---|---|---|
| | **Dice↑** | **ASSD(mm)↓** | **Dice↑** | **ASSD(mm)↓** |
| *State-of-the-Art Methods* | | | | |
| U-Net (Ronneberger et al., 2015) | 0.67 ± 0.17 | 6.11 ± 4.94 | 0.63 ± 0.21 | 8.06 ± 7.68 |
| Swin-UNETR (Hatamizadeh et al., 2021) | 0.69 ± 0.12 | 5.68 ± 4.53 | 0.65 ± 0.19 | 7.85 ± 7.01 |
| SAM-Med2D (Cheng et al., 2023) | 0.67 ± 0.15 | 5.81 ± 5.10 | 0.62 ± 0.18 | 7.16 ± 5.82 |
| ResUNet (Diakogiannis et al., 2020) | 0.71 ± 0.16 | 4.64 ± 3.91 | 0.69 ± 0.17 | 7.56 ± 7.11 |
| nnU-Net (Isensee et al., 2021) | 0.74 ± 0.14 | 5.11 ± 4.36 | 0.70 ± 0.14 | 6.53 ± 5.20 |
| ViT (Dosovitskiy et al., 2020) | 0.75 ± 0.13 | 4.54 ± 3.16 | 0.72 ± 0.11 | 6.06 ± 5.47 |
| Jain et al. (Jain et al., 2024) | 0.80 ± 0.10 | 4.32 ± 3.83 | 0.74 ± 0.13 | 5.49 ± 4.32 |
| *Ablation Study* | | | | |
| Cross-Att | 0.82 ± 0.10 | 4.08 ± 2.95 | 0.75 ± 0.11 | 5.11 ± 4.18 |
| Contrastive | 0.83 ± 0.08 | 3.81 ± 3.06 | 0.78 ± 0.13 | 5.03 ± 4.55 |
| **VocSegMRI** | **0.85 ± 0.06** | **3.17 ± 2.52** | **0.80 ± 0.10** | **4.89 ± 4.07** |

Cross-Att: cross-attention only. Contrastive: contrastive learning only.

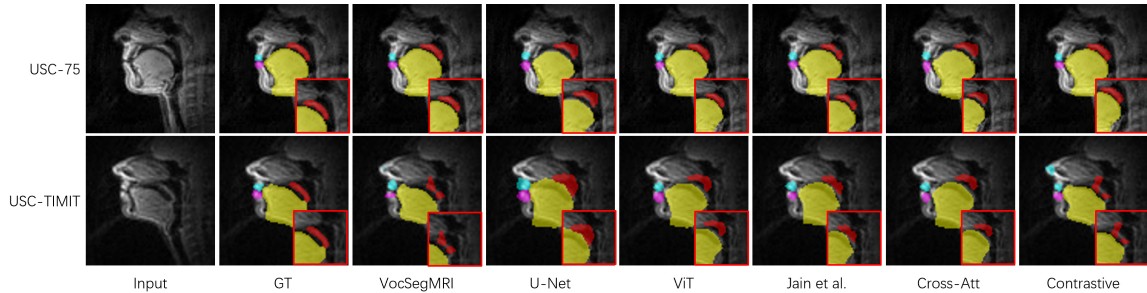

Figure 2: Qualitative segmentation results on one representative frame from USC-75 (top) and USC-TIMIT (bottom). Red insets zoom in on the velum region, one of the most challenging structures to segment accurately.

## 4. Discussion & Conclusion

VocSegMRI integrates visual, acoustic, and phonological information via cross-attention fusion and contrastive supervision, reaching a Dice of **0.85** and ASSD of **3.17 mm** on USC-75, representing a 6.25% Dice improvement and 1.15 mm ASSD reduction over the state-of-the-art multimodal baseline Jain et al. Cross-attention enables the visual encoder to selectively attend to complementary acoustic and phonological context, while contrastive supervision strengthens cross-modal alignment for more discriminative representations. Zero-shot transfer to USC-TIMIT further confirms cross-dataset generalizability. Nevertheless, future work will address temporal modeling across frames and coarticulation-aware phonological representations to further improve segmentation consistency.

## Acknowledgments

The authors acknowledge HPC resources provided by NHR@FAU of Friedrich-Alexander-Universität Erlangen-Nürnberg, funded by the German Research Foundation (DFG).

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

## Appendix A. Datasets

**USC-75** We utilized data from five participants (one male and four females) from the USC-75 dataset for training and in-domain evaluation. Each participant was instructed to read the *Rainbow* and *North Wind and the Sun* passages (Garofolo et al., 1993; Darley et al., 1975). The real-time MRI data were acquired at 83.28 fps with 2.4 mm in-plane resolution ($84 \times 84$ pixels) and 6 mm slice thickness (Lim et al., 2021). Imaging used a 1.5 T GE Signa Excite scanner; synchronized audio was recorded at 20 kHz via a fiber-optic microphone, hardware-locked to the scanner clock and denoised. The binary segmentation masks were manually annotated and further reviewed by a speech therapist expert, with recognized expertise in the field. Phonemes were obtained from the outputs of a phonological classifier, aligned with the corresponding phoneme labels from the audio stream, and subsequently refined through manual correction (Liu et al., 2025; Arias-Vergara et al., 2024).

**USC-TIMIT** This dataset comprises recordings from 10 native American English speakers (5M, 5F), each producing 460 phonetically balanced sentences, acquired at 23.18 fps with 2.9 mm isotropic resolution ($68 \times 68$ pixels) using a 1.5 T scanner. Synchronized speech was simultaneously recorded using a fiber-optic microphone. Articulator contours were extracted using the semi-automatic segmentation tool provided with the dataset.

## Appendix B. Implementation Details

We use a pretrained ViT-Base (`google/vit-base-patch16-224-in21k`) as the visual encoder and a pretrained WavLM-Base (`microsoft/wavlm-base-plus`) as the audio encoder, both with hidden dimension $D = 768$. Patch tokens (excluding the [CLS] token) from layers $\{4, 8, 12\}$ of the visual encoder are fused via learnable $1 \times 1$ convolutions to obtain multi-scale local features. A learnable 2-D positional embedding is added to the fused patch token map prior to the decoder, encoding spatial position information across the $14 \times 14$ patch grid. The Transformer decoder consists of 6 layers, the refined tokens are upsampled through two convolutional blocks with bilinear interpolation, followed by a $1 \times 1$ convolution yielding class logits for four articulator classes. The model is trained end-to-end with the following combined loss:

$$\mathcal{L} = \mathcal{L}_{\text{CE}} + \mathcal{L}_{\text{Dice}} + \lambda \cdot \mathcal{L}_{\text{contrast}} \tag{1}$$

where $\mathcal{L}_{\text{CE}}$ and $\mathcal{L}_{\text{Dice}}$ denote cross-entropy and Dice losses weighted equally, $\mathcal{L}_{\text{contrast}}$ is the contrastive loss, and $\lambda = 0.1$ controls its contribution. The source code will be released upon acceptance.

## Appendix C. Additional Results

Following Jain et al. (Jain et al., 2024), we implement a concat fusion baseline that directly concatenates encoder outputs for each modality. As shown in Tab. 2, incorporating audio or phonological features individually over the video-only baseline leads to moderate Dice improvements, with the best concat performance achieved using all three modalities. Notably, phonological features alone yield a smaller gain than audio signals, likely due to the difficulty of learning meaningful mappings between discrete phonological labels and continuous visual articulator regions. Nevertheless, all concat configurations are outperformed by VocSegMRI, confirming that cross-attention fusion and contrastive alignment provide more effective cross-modal integration than simple feature concatenation.

Table 2: Concat fusion ablation on USC-75.

| Modalities | Dice↑ | ASSD(mm)↓ |
|---|---|---|
| Video only (ViT) | $0.75 \pm 0.13$ | $4.54 \pm 3.16$ |
| Video + Audio | $0.80 \pm 0.10$ | $4.32 \pm 3.83$ |
| Video + Phonological | $0.79 \pm 0.09$ | $3.94 \pm 2.50$ |
| Video + Audio + Phonological | $0.81 \pm 0.10$ | $3.66 \pm 3.71$ |
| **VocSegMRI** | $\mathbf{0.85 \pm 0.06}$ | $\mathbf{3.17 \pm 2.52}$ |

