# OpenReview forum: "VocSegMRI: Multimodal Learning for Precise Vocal Tract Segmentation in Real-time MRI"
_MIDL.io/2026/Short_Papers — MIDL 2026 - Short Papers Poster_

### Official Review · Reviewer_PxE5 · 2026-05-02
**review for vocsegMRI**

**Rating:** 4
**Confidence:** 4

**Review:**

The paper provides a high-quality approach to a specialized problem in medical imaging: the segmentation of vocal tract structures during continuous speech. The clarity of the methodology is excellent, particularly the description of the cross-attention fusion mechanism between the ViT-based visual encoder and the WavLM-based audio encoder. Originality is found in the specific integration of phonological embeddings—voicing, place, and manner of articulation—which reflect the underlying physical constraints of speech production.  The results are significant, showing that the model successfully addresses the low signal-to-noise ratio inherent in rtMRI by leveraging complementary acoustic context. The use of a contrastive loss term $L_{contrast}$ is well-justified for aligning visual and audio-phonological tokens in a shared latent space. The technical depth is appropriate for MIDL, though the model currently operates on a frame-by-frame basis

**Summary:**

The authors introduce VocSegMRI, a deep learning framework that addresses the challenges of low-contrast articulatory segmentation in rtMRI by integrating synchronized audio and phonological features (voicing, place, and manner of articulation) with visual data. The architecture utilizes a ViT encoder for image features, a WavLM encoder for audio, and a Transformer decoder for cross-modal fusion, further supervised by a dual-level contrastive learning objective. Evaluated on the USC-75 dataset with zero-shot validation on the USC-TIMIT dataset, the model achieves a Dice score of 0.85, significantly outperforming unimodal models and previous multimodal state-of-the-art methods. This work is significant for its ability to leverage speech-related context to resolve visual ambiguities in medical imaging, particularly in challenging regions like the velum.

**Strengths:**

The main strength of VocSegMRI is its sophisticated integration of domain-specific knowledge—speech acoustics and phonological categories—into a computer vision task. By using a Transformer decoder for cross-attention, the model dynamically weighs the importance of audio cues relative to visual data. The inclusion of a contrastive learning head further ensures that the latent representations across modalities are well-aligned. The robust performance in zero-shot settings on the USC-TIMIT dataset is particularly impressive, suggesting the model captures fundamental articulatory-acoustic mappings.

**Weaknesses:**

The paper's primary weakness is the absence of temporal context; speech is inherently a temporal process, and segmenting frame-by-frame ignores the continuity of movement. Additionally, the reliance on high-level phonological features (voicing, place, manner) suggests that the model's performance might degrade if the input speech is pathological or if the phoneme alignment is noisy. The sample size of the dataset is also a limitation, as it may not fully represent the diversity of human vocal tract anatomy or different speech rates and styles.

**Justification Of Rating:**

The paper is a solid contribution to the MIDL community, specifically for articulatory imaging. It shows a clear improvement over existing methods ($Dice = 0.85$ vs $0.80$ for the previous SOTA). The methodology is modern and the validation includes cross-dataset testing, which is a high standard for short papers. While it lacks temporal modeling, the performance gains from the proposed multimodal fusion and contrastive alignment are significant enough to warrant acceptance.

---

### Decision · Program_Chairs · 2026-05-08

Accept (Poster)